# Comparison of the Single-Cell Immune Landscape of Testudines from Different Habitats

**DOI:** 10.3390/cells11244023

**Published:** 2022-12-12

**Authors:** Rui Guo, Guangwei Ma, Xiaofei Zhai, Haitao Shi, Jichao Wang

**Affiliations:** Ministry of Education Key Laboratory for Ecology of Tropical Islands, Key Laboratory of Tropical Animal and Plant Ecology of Hainan Province, College of Life Sciences, Hainan Normal University, Haikou 571158, China

**Keywords:** peripheral immune cell, single-cell RNA-sequencing, testudines

## Abstract

Testudines, also known as living fossils, have evolved diversely and comprise many species that occupy a variety of ecological niches. However, the immune adaptation of testudines to the different ecological niches remains poorly understood. This study compared the composition, function, and differentiation trajectories of peripheral immune cells in testudines (*Chelonia mydas*, *Trachemys scripta elegans*, *Chelonoidis carbonaria*, and *Pelodiscus sinensis*) from different habitats using the single-cell RNA sequencing (scRNA-seq) technique. The results showed that *T. scripta elegans*, which inhabits freshwater and brackish environments, had the most complex composition of peripheral immune cells, with 11 distinct immune cell subsets identified in total. The sea turtle *C. mydas*, had the simplest composition of peripheral immune cells, with only 5 distinct immune cell clusters. Surprisingly, neither basophils were found in *C. mydas* nor T cells in *C. carbonaria*. Basophil subsets in peripheral blood were identified for the first time; two basophil subtypes (GATA2-high-basophils and GATA2-low-basophils) were observed in the peripheral blood of *T. scripta elegans*. In addition, ACKR4 cells, CD4 T cells, CD7 T cells, serotriflin cells, and ficolin cells were specifically identified in the peripheral blood of *T. scripta elegans*. Furthermore, LY6G6C cells, SPC24 cells, and NKT cells were specifically observed in *C. carbonaria*. Moreover, there were differences in the functional status and developmental trajectory of peripheral immune cells among the testudine species. The identification of specific features of peripheral immune cells in testudines from different habitats may enable elucidation of the adaptation mechanism of testudines to various ecological niches.

## 1. Introduction

Testudines are some of the Earth’s most instantly recognizable life forms. They are known as living fossils, with fossil records dating back over 200 million years [1,2], and form an evolutionarily unique and morphologically distinctive clade of vertebrates that are imperiled on a global scale [3,4]. Globally, at least half of the 356 species of turtle are endangered, which is the highest risk of extinction among vertebrates. Turtles, characterized by the presence of a bony carapace and plastron that houses both the pectoral and pelvic girdles, possess one of the most derived tetrapod morphologies known. Moreover, turtles have an essentially global distribution, spanning an ecologically diverse set of terrestrial, marine, and freshwater habitats.

The immune system, including innate immunity and adaptive immunity, is an effective weapon for the body to defend against invasion by pathogenic microorganisms. Innate immunity is the innate immune mechanism of the body that recognizes constant antigens via pattern recognition receptors on the surface of immune cells, such as dendritic cells, macrophages, and neutrophils. In contrast to innate immunity, the adaptive immune system acts on specific antigens and has memory. Adaptive immunity can be divided into B cell-mediated humoral immunity and T cell-mediated cellular immunity. The molecular mechanisms of animal adaptation to various ecological niches have become a hot topic in the field of ecology and evolution. Different ecological niches have been shown to drive the adaptive evolution of immune-related gene families in mammalian bats [5,6]. Testudines occupy a variety of ecological niches because of their habitat and dietary diversity. In addition, testudines have a hard shell, which protects their head, abdomen, limbs, and tail from external injuries. Therefore, the immune adaptation of testudines to various ecological niches has received increasing attention.

Peripheral immune cells mainly participate in immune responses in testudines. Although different methods, such as bioimaging and histochemical staining, have been used to study the classification of the blood cells of turtles [7,8,9]. All five peripheral immune cells were identified in most turtles, namely basophils, eosinophils, lymphocytes, heterophils, and monocytes using traditional techniques (histochemical staining techniques, microscopic, and submicroscopic) [8,9,10]. However, turtle granulocytes cannot be accurately identified using traditional methods. There is a lack of consensus on their classification, which markedly hinders our understanding of detailed immune functions. The advent of single-cell RNA sequencing (scRNA-seq) techniques has allowed researchers to study non-classified cells based solely on the mRNA expression pattern of each cell. In particular, scRNA-seq not only helps to identify cell types but also resolves cell states. This method has been used for both invertebrates and vertebrates, including shrimp [11,12], oysters [13], hydra [14], mice [15], and humans [16]. The identification of specific cell states provides a better understanding of cell function.

*Chelonia mydas* is the second largest species of sea turtles, and is common in shallow tropical and subtropical waters, as well as on coastline beaches [17]. *C. mydas* has a maximum lifespan of 75 years. It begins its life as an omnivore and gradually shifts to a more herbivorous diet [18]. *Trachemys scripta elegans*, a freshwater and brackish turtle [19], is remarkably adaptable to the environment and is listed as one of the 100 most threatening aliens in the world by the International Union for Conservation of Nature. *T. scripta elegans* can live for up to 30 years in the wild, and captive sliders can live for up to 41.3 years [20]. Juvenile *T. scripta elegans* is mostly carnivorous, while the mature ones are omnivorous [21]. *Chelonoidis carbonaria* can be found in rainforests, dry thorny forests, and savanna areas [22]. *C. carbonaria* feeds primarily on fruits during the wet season and on flowers during the dry season [23]. It reaches sexual maturity at approximately 5 years of age [22]. *Pelodiscus sinensis* is an aquatic animal that often inhabits the sedimentary freshwater areas of rivers, lakes, ponds, and reservoirs, where freshwater teems with fish and shrimp [24]. *P. sinensis* is carnivorous, with fish, shrimp, and molluscs being the staple food [25]. The lifespan of wild *P. sinensis* is over 60 years [26].

The present study compared the composition, function, and differentiation trajectory of peripheral immune cells from four testudines (*C. mydas*, *T. scripta elegans*, *C. carbonaria*, and *P. sinensis*) occupying different ecological niches using scRNA-seq techniques. This study contributes to a comprehensive understanding of peripheral blood immune dynamics of testudines adapted to various ecological niches.

## 2. Materials and Methods

### 2.1. Sample Collection

In the present study, *C. mydas*, *T. scripta elegans*, *C. carbonaria*, and *P. sinensis* occupying different ecological niches were selected as research objects. *C. mydas* individuals that inhabited seawater were collected from the sea turtle rescue station of Hainan Normal University. *T. scripta elegans*, *C. carbonaria*, and *P. sinensis* individuals were collected from Dongshan Turtle Farm, in Haikou, China. *T. scripta elegans* inhabited in freshwater, *C. carbonaria* inhabited land, and *P. sinensis* inhabited sedimentary freshwater. All testudines were apparently healthy and without any injuries or deformities. Individually, they were in excellent condition. The body weight of *C. mydas* was 43.1 ± 5.8 kg, and the length and width of the carapace were 74.2 ± 5.2 cm and 65.7 ± 6.0 cm, respectively (Appendix A). The average body weight of *T. scripta elegans* was 2.2 ± 0.3 kg, while the average length and carapace width were 24.5 ± 1.6 cm and 17.8 ± 1.2 cm, respectively (Appendix A). The body weight of *C. carbonaria* was 1.5 ± 0.1 kg, and the length and width of the carapace were 19.9 ± 0.7 cm and 12.9 ± 0.2 cm, respectively (Appendix A). The average body weight of *P. sinensis* was 1.6 ± 0.2 kg, and the average length and width of the carapace were 22.8 ± 0.7 cm and 19.2 ± 1.0 cm, respectively (Appendix A). Peripheral blood samples from adult *C. mydas* (n = 3), *T. scripta elegans* (n = 6), *C. carbonaria* (n = 3), and *P. sinensis* (n = 6) were collected from the jugular vein. The peripheral blood samples were stored on ice. All samples were subjected to cell isolation within 1 h of sampling.

### 2.2. Preparation of Single Cell Suspension

Peripheral blood (1 mL) was extracted into a 1.5 mL enzyme-free centrifuge tube containing a small amount of EDTA-K2 (2 mg) and incubated at 20 °C for 10 min. The serum plasma was discarded, and 1 mL of pre-cooled nucleated red cell lysate was added and incubated at 4 °C for 10 min. The suspension was then centrifuged at 3000 rpm for 10 min at 4 °C to discard the supernatant, following which, 0.1 mL of pre-cooled sterile enzyme-free PBS was added with gentle blowing to re-suspend the white precipitate at the bottom of the centrifuge tube. Cell concentration was measured using a cell counter or blood counting plate, and cell viability was calculated using trypan blue staining. An appropriate volume of pre-cooled, enzyme-free, sterile PBS diluted single-cell suspension was added to ensure that the final cell concentration was 10× Genomics. cDNA libraries were sequenced on the Illumina sequencing platform by Genedenovo Biotechnology Co., Ltd. (Guangzhou, China).

### 2.3. ScRNA-seq Data Processing and Analysis

The 10X Genomics Cell Ranger software (version 3.1.0, USA) was used to convert raw BCL files to FASTQ files for alignment and count quantification. The reads were then mapped to *C. mydas* (GCA_015237465.2 rCheMyd1.pri.v2), *T. scripta elegans* (GCA_013100865.1 CAS_Tse_1.0), *C. carbonaria* (GCA_003597395.1 ASM359739v1), and *P. sinensis* (GCA_000230535.1 PelSin_1.0) genome. The cell-by-gene matrices for each sample were individually imported into Seurat (version 3.1.1) for downstream analysis [27]. Cells with unusually high numbers of unique molecular identifiers (UMIs) (≥8000) or mitochondrial gene percentages (≥10%) were excluded. We also excluded cells with under 500 or over 4000 detected genes. Doublet gel beads-in-emulsion (GEMs) were also filtered out. After removing the unwanted cells from the dataset, we employed a global-scaling normalization method (“LogNormalize”) that normalizes the gene expression measurements of each cell by the total expression, multiplies this by a scale factor (10,000 by default), and log-transforms the results, to normalize the data. To minimize the effects of batch variability and behavioral conditions on clustering, we used Harmony [28], an algorithm that projects cells into a shared embedding in which cells are grouped by cell type rather than dataset-specific conditions, to aggregate all samples. The resulting integrated expression matrix was then scaled and dimensionally reduced using principal component analysis (PCA). The Seurat software, which implements a graph-based clustering approach, was used to cluster the cells. A resolution of 0.6 was chosen as the clustering parameter. For visualization of the clusters, a UMAP was generated using the same PCs. The log-normalized matrices were then loaded onto the SingleR R package for cell type annotation, which is based on correlating the gene expression of reference cell types with that of single cells.

Downstream analyses, including normalization, shared nearest-neighbor graph-based clustering, differential expression analysis, and visualization, were performed using the standard workflow provided by Seurat (version 3.1.1, Satija Lab, New York, NY, USA). The expression value of each gene in a given cluster was compared against that of the rest of the cells using the Wilcoxon rank-sum test in Seurat [29]. Significantly upregulated genes were identified using the criteria: log_2_ FC ≥ 0.36 and Bonferroni-adjusted *p* ≤ 0.01. Differentially expressed genes (DEGs) were subjected to the GO analysis to screen immune cell subsets and then to the KEGG analysis to illustrate the function signatures of the subsets. Developmental trajectories were inferred with the Monocle (version 2.10.1, Cole Trapnell and Davide Cacchiarelli, USA). Monocle reduced the space to one with two dimensions and ordered the cells (sigma = 0.001, lambda = NULL, param. gamma = 10, tol = 0.001) [30]. Once the cells were ordered, the trajectory was visualized in reduced dimensional space, and it had a tree-like structure with tips and branches. Monocle was used to identify genes that were differentially expressed between the groups of cells, and genes with similar trends in expressions, such as shared common biological functions and regulators, were grouped (Appendix A).

## 3. Results

### 3.1. Landscapes of Peripheral Immune Cells in Testudines

To characterize the peripheral blood immune cells, we applied the scRNA-seq technique to identify the immune cells isolated from the peripheral blood of ecologically diverse testudines. Following computational quality control and filtering using the Seurat package, all 16,578 high-quality cells were subjected to further analysis (Appendix A). There were 7222, 6888, 7222, and 4599 DEGs selected from peripheral immune cells of *C. mydas*, *T. scripta elegans*, *C. carbonaria*, and *P. sinensis*, respectively (Appendix A). Examination of canonical marker genes revealed the major cell populations involved in innate and adaptive immunity. Monocytes were identified using the marker genes *LGALS1*, *LY86*, *PLBD1*, *MYOF*, *BPI*, and *CSF1R* (Figure 1). Neutrophils were identified by the marker gene *CXCR2* (Figure 2). B cells were observed by the marker genes *LAMP3* and *PAX5* with upregulated *CD79A/B* (Figure 2 and Figure 3A–D), while T cells were inferred by the marker genes *BCL11B* and *CD3E* (Figure 2). The marker gene of basophils was *GATA2* (Figure 2). ACKR4-cell, LY6G6C-cell, and SPC24-cell lines were identified by upregulated *ACKR4*, *LY6G6C*, and *SPC24* genes, respectively (Figure 2). Serotriflin-cell type was inferred by the upregulated *SEROTRIFLIN* and *PET2* genes (Figure 2). Ficolin cells were identified by the upregulation of *FICOLIN* and *PET2* genes (Figure 2). NKT cells were identified by the upregulated *OGT*, *PRPF4B*, and *PNISR* genes (Figure 3C).

When comparing the composition of peripheral immune cells among testudines inhabiting different environments, we found differences between the different species. *T. scripta elegans* had the largest variety of immune cells (11 types), whereas *C. mydas* had the fewest types of immune cells (only 5 types). The immune cell types in the peripheral blood of *T. scripta elegans* included ACKR4 cells, B cells, CD4 T cells, CSF1R monocytes, GATA2-high-basophils, serotriflin cells, neutrophils, POF1B monocytes, ficolin cells, CD7 T cells, and GATA2-low-basophils (Figure 4B). Among them, ACKR4 cells, serotriflin cells, and ficolin cells were newly identified immune cell types in *T. scripta elegans* (Figure 4B). Only 5 types of immune cells were identified in the peripheral blood of *C. mydas*, including neutrophils, CSF1R monocytes, T cells, RAB3B monocytes, and B cells (Figure 4B). Moreover, 8 types of immune cells (neutrophils, LY6G6C cells, SPC24 cells, CSF1R monocytes, basophils, NKT cells, B cells, and GPRC5A monocytes) were identified in the peripheral blood of *C. carbonaria* (Figure 4B). Among these, LY6G6C cells, NKT cells, and SPC24 cells were newly discovered immune cell types in *C. carbonaria* (Figure 4B). Six types of immune cells (neutrophils, CSF1R monocytes, basophils, LARP6 monocytes, T cells, and B cells) were identified in the peripheral blood of *P. sinensis* (Figure 4B). Among the peripheral blood immune cells, neutrophils, CSF1R monocytes, and B cells were universal to all four testudines species. Surprisingly, no basophils were found in the peripheral blood of *C. mydas*, and no T cells were observed in the peripheral blood of *C. carbonaria*. Compared to other testudine species, the T cells and basophils in *T. scripta elegans* were classified into two subtypes: CD4 T and CD7 T cells, and GATA2-high-basophil and GATA2-low-basophil (Figure 4B). The innate immune cells of peripheral blood included neutrophils, monocytes, basophils, serotriflin cells, NKT cells, and ficolin cells, while B cells, T cells, ACKR4 cells, LY6G6C cells, and SPC24 cells represented the adaptive immune cells in the four species.

### 3.2. Innate Immune Function of Peripheral Blood in Testudines

The KEGG analysis was used to analyze the significantly enriched pathways in the upregulated genes, and the functional differences of each innate immune cell were compared among species. In testudines, monocytes were typically classified as CSF1R monocytes and other monocyte types (RAB3B monocytes, POF1B monocytes, GPRC5A monocytes, and LARP6 monocytes) (Figure 4B). Among monocytes, upregulated genes differed between subtypes and species (*C. mydas* CSF1R monocyte: *C1QC*, *ACOT11*, *ATP6V0D2*, *C1QB*, and *QPCT*; *T. scripta elegans* CSF1R monocyte: *OAF*, *LOC117873117*, *EMP1*, *FABP7*, and *HSD11B2*; *C. carbonaria* CSF1R monocyte: *LOC116832753*, *KCNA5*, *LOC116834952*, *ANO1*, and *CUNH11orf86*; *P. sinensis* CSF1R monocyte: *LOC106732254*, *FN1*, *LOC112545434*, *PROC*, and *RIN1*; *C. mydas* RAB3B monocyte: *LOC102948104*, *RAB3B*, *LOC119564253*, *HOXA7*, and *ST6GAL2*; *T. scripta elegans* POF1B monocyte: *LOC117884757*, *POF1B*, *LOC117874926*, *LOC117883046*, and *NECTIN1*; *C. carbonaria* GPRC5A monocyte: *LOC116824001*, *MYCN*, *LOC116827637*, *GPRC5A*, and *LOC116816276*; *P. sinensis* LARP6 monocyte: *LOC102446777*, *LARP6*, *DNASE2B*, *SLC6A15*, and *ERP29*) (Figure 3A–D). KEGG terms of specifically upregulated genes in CSF1R monocytes in these four testudines were enriched in biological pathways for oxidative phosphorylation, disease (Parkinson’s disease, non-alcoholic fatty liver disease, Huntington’s disease, Alzheimer’s disease, etc.), and lysosomes (Figure 5). Remarkably, CSF1R monocytes in *C. mydas*, *T. scripta elegans*, and *C. carbonaria* had phagosomes, and CSF1R monocytes in *C. mydas* showed endocytosis (Figure 5). The functions of other monocytes (RAB3B monocytes, POF1B monocytes, GPRC5A monocytes, and LARP6 monocytes) were similar to those of CSF1R monocytes (Figure 5).

Neutrophils were the most common cell type found in the peripheral blood of testudines, and constituted a major part of the innate immune system. They were classified as granulocytes along with basophils. The upregulated genes of neutrophils differed among species (*C. mydas*: *LOC114022268*, *LOC119567953*, *LOC119566555*, *LOC102942826*, and *CXCR2*; *T. scripta elegans*: *ACOD1*, *LOC117873143*, *LOC117885978*, *LOC117869779*, and *CSF3R*; *C. carbonaria*: *LOC116820014*, *CXCR2*, *LOC116838624*, *TRIM39*, and *LOC116829384*; *P. sinensis*: *LOC102450310*, *LOC102454433*, *TF*, *CUNH11orf96*, and *LOC102456901*) (Figure 3A–D). According to KEGG analysis, upregulated genes of neutrophil in the four testudines were enriched in biological pathways associated with viral and bacterial infections, cancer, fluid shear stress, and atherosclerosis (Figure 5). Except for *C. carbonaria*, neutrophils from other testudines showed endocytosis and chemokine signaling pathways (Figure 5). Fc gamma R-mediated phagocytosis, TNF signaling pathway, and fat digestion and absorption were specifically enriched in the neutrophils of *C. mydas* (Figure 5). The oxytocin signaling pathway and dopaminergic synapses, as well as viral and bacterial infection pathways, were specifically enriched in the neutrophils of *T. scripta elegans* (Figure 5). Pentose phosphate, endoplasmic reticulum protein processing, NAFLD, sphingolipid signaling, and mTOR signaling pathways were specifically enriched in the neutrophils of *P. sinensis* (Figure 5). Platelet activation, parathyroid hormone synthesis, secretion and action, cell adhesion molecule, hematopoietic cell lineage, complement, and coagulation cascade, and Jak-STAT signaling pathway were specifically enriched in the neutrophils of *C. carbonaria* (Figure 5). The IL-17 signaling pathway was enriched only in the neutrophils of *C. mydas* and *C. carbonaria* (Figure 5).

Basophils comprised a rare subset of granulocytes in testudines. Similar to neutrophils, upregulated basophil genes differed among species (GATA2-high-basophils: *LOC117885767*, *LOC117885781*, *GATA2*, *TNFAIP8L3*, and *GPC1*; GATA2-low-basophils: *LOC117886291*, *LIF*, *LOC117872690*, *LOC117883374*, and *TGFB2*; *C. carbonaria*: *TGFB2*, *LOC116816807*, *SLC38A11*, *CST7*, and *RAB27B*; *P. sinensis*: *LOC102456708*, *LOC102455777*, *SYTL2*, *LOC102454962*, and *LOC106732001*) (Figure 3A–D). KEGG-enriched pathways were also different (Figure 5). In *T. scripta elegans*, GATA2-high-basophils were related to pathways involved in regulating T cells, cancer, leishmaniasis, apoptosis, and mitophagy, whereas GATA2-low-basophils were related to pathways in cancer, T cell differentiation, and defense against parasites (Figure 5). The basophils of *C. carbonaria* were related to pathways in cancer, chemokine signaling, autophagy, apoptosis, and phagocytosis (Figure 5), while those of *P. sinensis* were related to the pathways involved in disease, autophagy, and cancer (Figure 5). Thus, peripheral blood basophils in testudines were involved in the defense against parasites and played a role in allergic reactions.

As for the newly discovered innate immune cell types, *LOC117875385*, *LOC117886016*, *PLK3*, *LOC117873755*, and *LOC117869907* were specifically upregulated in the serotriflin-cell type of *T. scripta elegans*, and were enriched in pathways involved in viral and bacterial infections, parasitosis, cancer, endocytosis, and phagocytosis; *LOC117867189*, *LOC117868675*, *LOC117873455*, *HRH4*, and *LOC117868150* were highly expressed in the ficolin cells of *T. scripta elegans*, and were enriched in pathways related to endocytosis, disease, pathogenic infection, leukocyte transendothelial migration, chemokine signaling, necroptosis, and oxidative phosphorylation; *PITPNM2*, *OGT*, *PRPF4B*, *LOC116835191*, and *PNISR* were highly expressed in the NKT cells of *C. carbonaria* and were enriched in pathways related to virus infection, cancer, B cell receptor signaling, natural killer cell-mediated cytotoxicity, and endocytosis (Figure 3 and Figure 6).

### 3.3. Adaptive Immune Function of Peripheral Blood in Testudines

B cells are essential components of the adaptive immune system. The upregulated genes in B cells differed among the testudine species. The upregulated genes in the B cells of the four species are as follows: *LAMP3*, *CD79B*, *LOC119566643*, *CD209B*, and *PAX5* in *C. mydas*; *CD79B*, *LOC117870462*, *EBF1*, *TNFRSF13B*, and *JCHAIN* in *T. scripta elegans*; *PAX5*, *LOC116833347*, *CD79A*, *CD79B*, and *PLAC8* in *C. carbonaria*; *TNFRSF13C*, *PAX5*, *LOC112543738*, *LOC102451180*, and *COL14A1* in *P. sinensis* (Figure 3A–D). KEGG terms of specifically upregulated genes in B cells were similar across species and were enriched in pathways in B cell development and activation, cancer, B cell receptor signaling, immunologic network, and phagocytosis (Figure 5).

T cells are essential for maintaining the immune balance by supporting the protection from foreign pathogens, suppressing excessive immune reactions, and preventing autoimmunity. The upregulated genes in T cells also differed among the testudine species. The upregulated genes were as follows: *LOC102932316*, *CD3E*, *LOC102935423*, *CD3D*, and *ETS1* in the T cells of *C. mydas*; *LOC117870832*, *LOC117885267*, *CD5*, *LOC117887934*, and *TCF7* in the CD4 T cells of *T. scripta elegans*; *LOC117887273*, *LOC117872867*, *LOC117885533*, *CD7*, and *EOMES* in the CD7 T cells of *T. scripta elegans*; *CD226*, *UBASH3A*, *TCF7*, *BCL11B*, and *LOC106732215* in the T cells of *P. sinensis* (Figure 3A,B,D). KEGG terms of specifically upregulated genes in T cells were similar across species, and were enriched in pathways of the primary immunodeficiency, inflammatory diseases, T cell differentiation, and T cell receptor signaling (Figure 5). Surprisingly, CD4 T cells and CD7 T cells showed similar functions in *T. scripta elegans* (Figure 5).

In particular, ACKR4 cells in *T. scripta elegans*, and LY6G6C cells and SPC24 cells in *C. carbonaria*, were first found to be lymphocytes distinct from common T cells and B cells. *ACKR4*, *SYNM*, *SOX6*, *AQP1*, and *SELENOP* were found highly expressed in the ACKR4 cells of *T. scripta elegans* (Figure 3B). KEGG terms of the specifically expressed genes in ACKR4 cells included oxidative phosphorylation, disease, viral infection, autophagy, and mitophagy (Figure 6). *LY6G6C*, *LOC116838262*, *ALS2CL*, *LOC116821286*, and *LOC116829202* were upregulated in the LY6G6C cells of *C. carbonaria* (Figure 4B). KEGG terms of the upregulated genes in LY6G6C cells included viral infection pathways (Figure 6). High expression of *SPC24*, *NES*, *LOC116828075*, *CLSTN3*, and *PLAC8* was observed in the SPC24 cells of *C. carbonaria* (Figure 3C), which were enriched in pathways involved in viral and bacterial infections, disease, and leukocyte transendothelial migration (Figure 6).

### 3.4. Developmental Trajectories of Peripheral Immune Cells in Testudines

The developmental trajectory of peripheral immune cells was analyzed by performing pseudotime analysis on our scRNA-seq data. The differentiation trajectories of peripheral immune cells differed among species. Two lineages (neutrophil and monocyte/lymphoid lineages) were observed in the peripheral immune cells of *C. mydas* (Figure 7A), while those of *T. scripta elegans*, *C. carbonaria*, and *P. sinensis* showed three lineages (lymphocyte-like/basophil, monocyte/lymphoid, and neutrophil lineages) (Figure 7B−D). In *T. scripta elegans*, ACKR4 cells belonged to a separate lineage (lymphocyte-like lineage); basophils, CD4 T cells, CD7 T cells, B cells, and monocytes belonged to the monocyte/lymphoid lineage; and neutrophils, ficolin cells, and serotriflin cells belonged to the neutrophil lineage (Figure 7B). In *C. carbonaria*, LY6G6C cells and SPC24 cells from the same lineage, namely the lymphocyte-like cell lineage; neutrophils and NKT cells belonged to neutrophil lineage; and basophils, B cells, and monocytes belonged to the monocyte/lymphoid lineage (Figure 7C). In *P. sinensis*, neutrophils and basophils derived from neutrophil and basophil lineages, respectively, while monocytes, B cells, and T cells diverged from a single trajectory (monocyte/lymphoid lineage) (Figure 7D).

## 4. Discussion

In summary, we established a more comprehensive single-cell transcriptomic atlas of peripheral immune cells in testudines from different habitats. We integrated the complex information on immune cells, including cell composition, functional status, and developmental trajectory.

In this study, the peripheral immune cells of testudines were identified using scRNA-seq technique for the first time. Previous studies using traditional light microscopy methods have identified five types of peripheral immune cells in testudines (*Testudo graeca*, *Emys orbicularis*, *Mauremys capsica*, *T. scripta scripta*, *C. mydas*, *Caretta caretta*, *C. carbonaria*, and *Lepidochelys olivacea*), namely, basophils, eosinophils, lymphocytes, heterophils, and monocytes [7,8,9,10,31,32]. However, in *Agrionemys horsfieldi*, seven types of peripheral immune cells were identified by light microscopy methods: lymphocytes, monocytes, heterophils, eosinophils, azurophils, basophils, and toxic heterophils [33]. The rapid development of scRNA-seq technology provides a powerful tool and unprecedented opportunity to define cell taxonomy, track differentiation, and uncover transcriptional networks at single-cell resolution for any given isolatable heterogeneous cell population. At present, research on peripheral blood immune cells by scRNA-seq technology has mainly been applied in the medical field [16,34,35], as well as to research immune defenses in other species such as the molluscan *Crassostrea hongkongensis* [13] and mammalian mice [36]. 

Compared with traditional classification methods, the scRNA-seq technique enabled the identification of more peripheral immune cell types in testudines. Moreover, ACKR4 cells, serotriflin cells, ficolin cells, LY6G6C cells, SPC24 cells, and NKT cells were found in peripheral blood for the first time. These cells could not be identified by canonical cell markers; therefore, they were named based on their upregulated genes. Other immune cells were identified using canonical cell markers combined with upregulated genes. Among the four testudine species, neutrophils, monocytes, basophils, serotriflin cells, NKT cells, and ficolin cells were innate immune cells, while B cells, T cells, ACKR4 cells, LY6G6C cells, and SPC24 cells were adaptive immune cells.

Monocytes, as circulating white blood cells, patrol the body and populate specific tissues in response to environmental stimuli. In human peripheral blood, monocytes can be divided into different subsets based on the expression of the surface markers *CD14* and *CD16* [37]. In contrast to the findings of previous studies, *CD14* and *CD16* were not expressed in the peripheral blood monocytes of the testudines, while six marker genes, *LGALS1*, *LY86*, *PLBD1*, *MYOF*, *BPI*, and *CSF1R*, were highly expressed and enriched in response to positive regulation of stimulus and stress, leukocyte chemotaxis, and migration (Appendix A). Previous studies have shown that *LGALS1*, *LY86*, *PLBD1*, *BPI*, and *CSF1R* were expressed in monocytes [38,39,40,41,42]. *MYOF* was observed to be expressed in monocytes for the first time; it was mainly involved in biological processes of response to temperature stimulus and syncytium formation (Appendix A). Based on these marker genes, we identified two distinct monocytes in the peripheral blood cells of *C. mydas* (CSF1R monocyte and RAB3B monocyte), *T. scripta elegans* (CSF1R monocyte and POF1B monocyte), *C. carbonaria* (CSF1R monocyte and GPRC5A monocyte), and *P. sinensis* (CSF1R monocyte and LARP6 monocyte).

Peripheral blood monocytes in testudines had several interesting characteristics. Two monocyte subtypes were identified in each testudine species, and upregulated genes differed among the monocyte subtypes. However, according to GO and KEGG analyses, the functions of the monocyte subtypes were not significantly different from each other and were similar to those of inflammatory monocytes with pro-inflammatory and antimicrobial functions in mice [43]. The differences in monocyte subtypes in the peripheral blood of testudines need to be further studied.

Surprisingly, neutrophils and basophils were the only granulocytes identified by scRNA-seq technology in the testudines; no eosinophils were found. Neutrophils and basophils were identified using the *CXCR2* and *GATA2* marker genes, respectively. *CXCR2* served as a cognate neutrophil receptor that drove neutrophil migration and was a marker for mature neutrophils [44]. GATA2 was required to maintain Fcer1a mRNA and FcεRIa protein expression in basophils [45]. Interestingly, functional differences in neutrophils and basophils were observed among the testudine species. Compared with other testudine species, neutrophils from *T. scripta elegans* typically responded to acute inflammation, such as viral or bacterial infection or cancers. Meanwhile, basophil subsets in the peripheral blood were observed for the first time, with two basophil subtypes observed in the peripheral blood of *T. scripta elegans*. Surprisingly, basophils were not detected in the peripheral blood of *C. mydas*. Basophils contributed to Th2 immunity at various levels [46]. However, in *T. scripta elegans*, basophils were involved in Th1, Th2, and Th17 immunity at various levels. In addition, the basophils of *T. scripta elegans* and *C. carbonaria* regulated the B cell receptor signaling pathway. The current understanding of the interaction between B cells and basophils indicated that basophils sustained malignant clone expansion in B cell lymphoproliferative diseases through immune regulatory functions [47]. Therefore, we predicted that the interactions between basophils and B/T cells could be a connection between innate and adaptive immunity in the peripheral blood of testudines.

NKT cells were unique unconventional T cells that shared properties of T cells and NK cells, they had characteristics of both innate and adaptive immune cells, and had potent immunoregulatory roles in tumor immunity, autoimmunity, and infectious diseases [48,49]. There were two main NKT cell subsets in humans and mice [49]. However, in this study, NKT cells were identified only in the terrestrial *C. carbonaria*, and only one cell subtype was observed. NKT cells of the peripheral blood in *C. carbonaria* were identified by *PITPNM2*, *OGT*, *PRPF4B*, *PNISR*, and *CD3E* genes, which had immunoregulatory roles in virus infection and cancer. Moreover, NKT cells diverged from the neutrophil lineage. For this reason, NKT cells were classified as innate immune cells in the present study.

B cells, T cells, ACKR4 cells, LY6G6C cells, and SPC24 cells belonged to adaptive immune responses. Surprisingly, three lymphocyte-like cell subsets (ACKR4 cells, LY6G6C cells, and SPC24 cells) were identified in the peripheral blood of *T. scripta elegans* and *C. carbonaria*. All mature B cell types depended on *PAX5* for their differentiation and function, and conditional loss of *PAX5* allows mature B cells from peripheral lymphoid organs to develop into functional T cells in the thymus via dedifferentiation into uncommitted progenitors in the bone marrow [50]. The present study is the first to report the marker gene *LAMP3* in B cells. This study identified B cells using marker genes *PAX5* and *LAMP3*. No differences were observed in the classification and function of B cells among the testudine species. In contrast to B cells, the classification of peripheral blood T cells varied greatly among the four species. T cells were identified using the marker genes *BCL11B* and *CD3E*. Surprisingly, no T cells were identified in the peripheral blood of *C. carbonaria*. The peripheral blood of *T. scripta elegans* possessed two T cell subtypes, while only one T cell subtype was observed in the peripheral blood of *C. mydas* and *P. sinensis*. In contrast to the T cell subtypes (CD4 T cells and CD8 T cells) of mammalian peripheral blood cells [16], T cells were classified as CD4 T cells and CD7 T cells in the peripheral blood of *T. scripta elegans*. Additionally, the NF-kappa B signaling pathway was specifically observed in the CD4 T cells.

According to the developmental trajectory of peripheral immune cells, neutrophil lineages, monocyte/lymphocyte lineages, and lymphocyte-like lineages were found in testudines. In human peripheral blood, neutrophils and monocytes diverged into the same trajectory (neutrophil/monocyte lineage), and B cells, T cells, and NK cells diverged into the lymphocyte lineage [16]. However, the present study showed that monocytes, T cells, and B cells diverged into monocyte/lymphocyte lineages; neutrophils, serotriflin cells, NKT cells, and ficolin cells diverged into neutrophil lineages; and ACKR4 cells, LY6G6C cells, and SPC24 cells diverged into lymphocyte-like lineages. Therefore, serotriflin cells, ficolin cells, and NKT cells should be granulocytes and participate in innate immunity, while ACKR4 cells, LY6G6C cells, and SPC24 cells should be similar to lymphocytes and participate in adaptive immunity. Unlike human peripheral blood, neutrophils and monocytes in testudines diverged into different trajectories (neutrophil and monocyte/lymphoid lineages), while monocytes, T cells, and B cells diverged into the same trajectory (monocyte/lymphocyte lineages). In addition, the differentiation trajectories of peripheral immune cells differed among species. These results illustrated that peripheral immune cells were heterogeneous and complex in testudines, further experiments are required to verify. Moreover, only peripheral blood immune cells were collected in this study, blood immune cells at different differentiation stages were not obtained, which could not completely explain the developmental trajectory of immune cells in testudines. Later, blood samples at different development stages should be collected for further analysis.

As described, the present study established a more comprehensive single-cell transcriptomic atlas of peripheral immune cells in testudines occupying different ecological environments. The present results showed that the composition of peripheral immune cells differed among testudines from different habitats. *T. scripta elegans* from freshwater and brackish environment had the most complex composition of peripheral immune cells, with 11 distinct immune cell clusters identified in total. *C. mydas*, which inhabits shallow tropical and subtropical waters as well as coastline beaches, had the simplest composition of peripheral immune cells, with only five distinct immune cell clusters. The numbers of immune cell clusters in the terrestrial *C. carbonaria* and *P. sinensis* from sedimentary freshwater areas were between those of *T. scripta elegans* and *C. mydas*. Eight distinct immune cell clusters were identified in *C. carbonaria*, while six distinct immune cell clusters were identified in *P. sinensis*. Interestingly, the peripheral immune cells were the most diverse in *T. scripta elegans*, which may also be one reason why *T. scripta elegans* is remarkably adaptable to the environment, as one of the 100 most threatening aliens in the world. These results illustrated that the more complex environment of testudines inhabited, the more diverse the peripheral immune cells were. Because different ecological niches could drive the adaptive evolution of immune-related gene families [5,6]. So different ecological niches may drive the adaptive evolution of peripheral immune cells in testudines.

Our study illustrated the peripheral immune landscape of testudines from the ocean, freshwater, sedimentary freshwater, rainforest, and savanna areas. These findings indicated that the more complex the habitat environment of testudines, the more abundant the peripheral immune cell population. Further functional studies using appropriate in vitro or in vivo models are necessary to clarify immune differences among testudines from different habitats, which may provide new insights into the adaptive evolution of peripheral immune cells in testudines.

## Figures and Tables

**Figure 1 cells-11-04023-f001:**
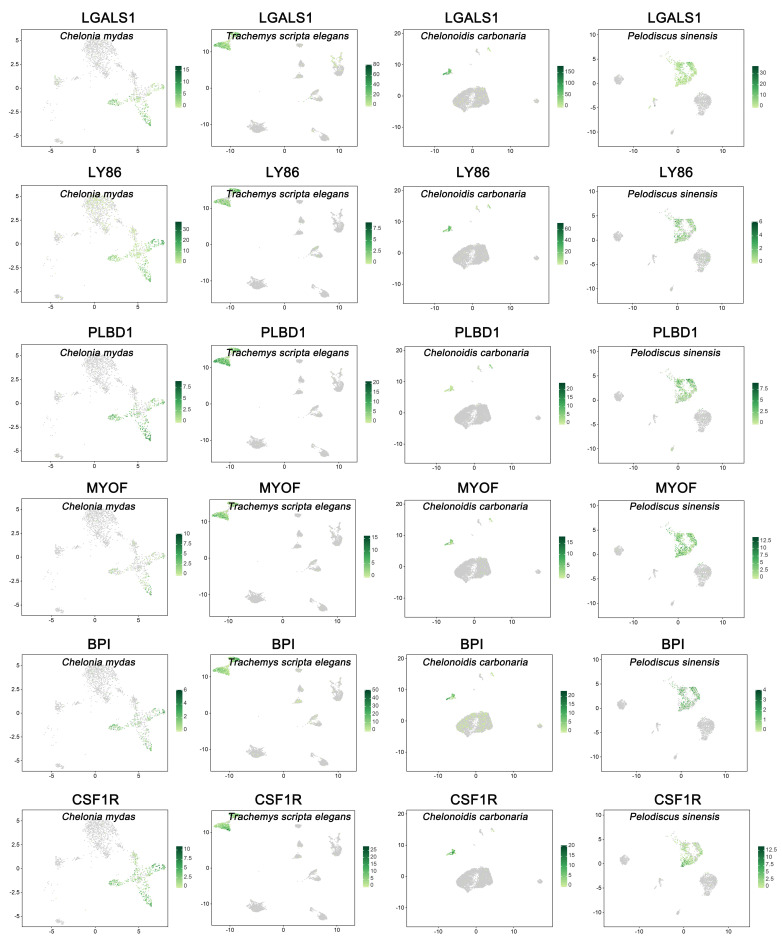
Canonical cell markers of monocytes were used to label clusters by cell identity as represented in the UMAP plot.

**Figure 2 cells-11-04023-f002:**
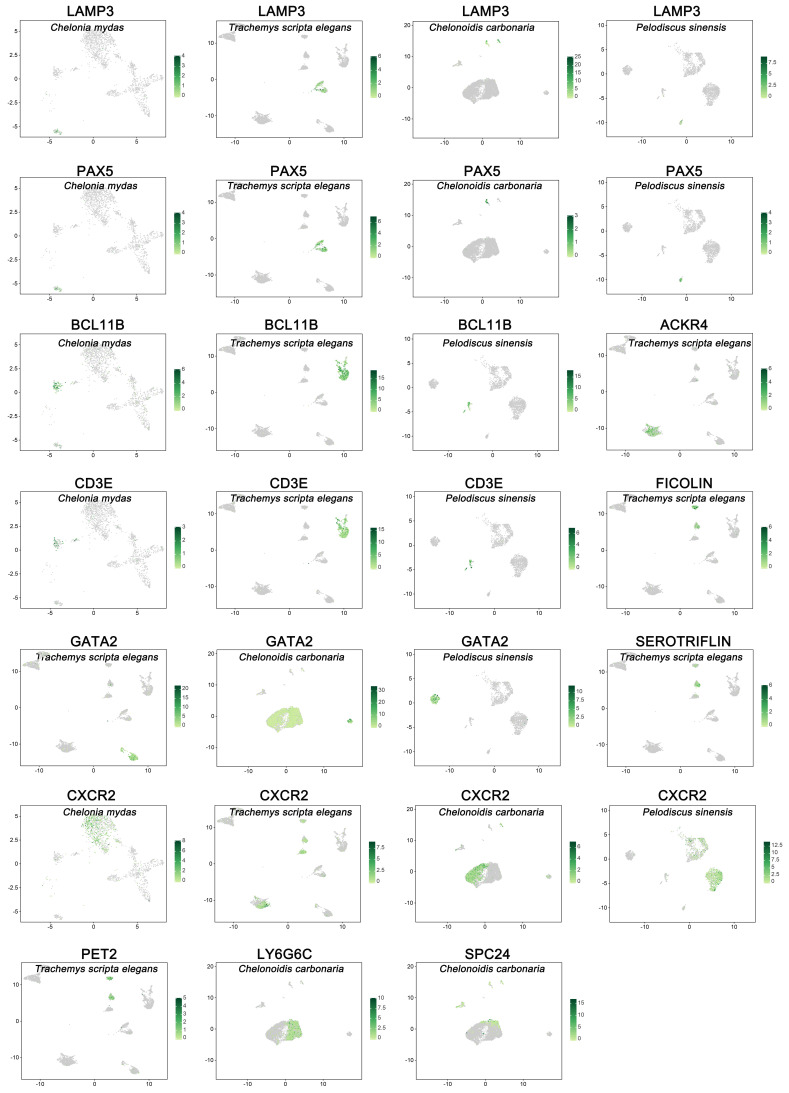
Canonical cell markers were used to label clusters by cell identity as represented in the UMAP plot.

**Figure 3 cells-11-04023-f003:**
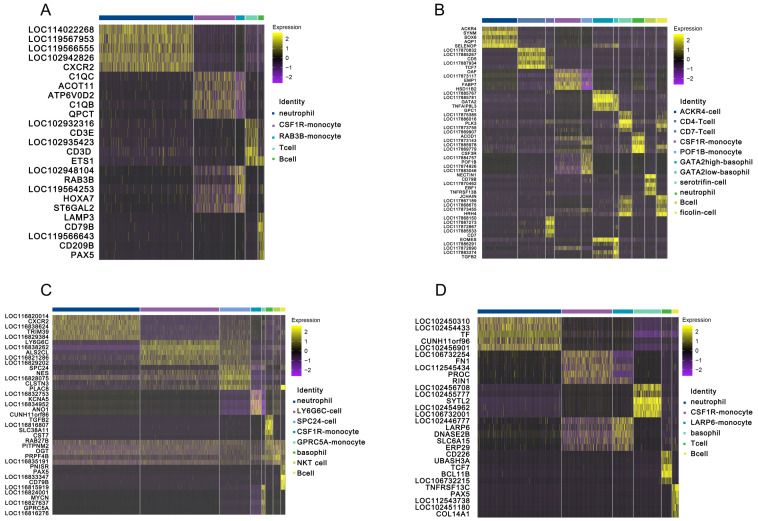
Heat map of upregulated genes in the peripheral blood immune cells of testudines. (**A**) *C. mydas*; (**B**) *T. scripta elegans*; (**C**) *C. carbonaria*; and (**D**) *P. sinensis*.

**Figure 4 cells-11-04023-f004:**
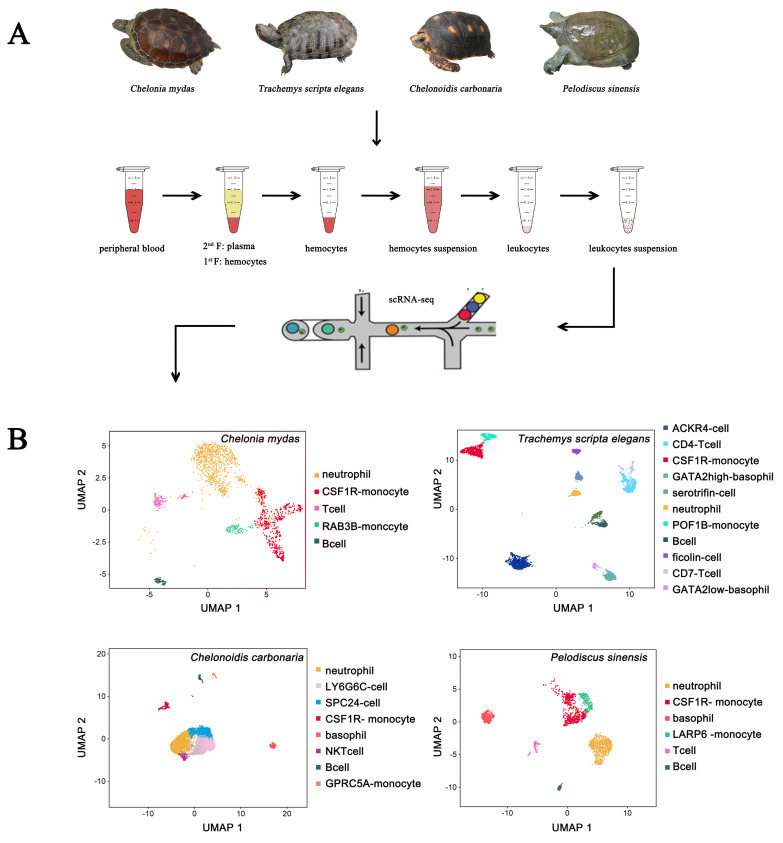
Transcriptome reference of peripheral immune cells. (**A**) A schematic workflow of the experimental design. (**B**) UMAP showing the distribution of peripheral immune cell types among the four testudines.

**Figure 5 cells-11-04023-f005:**
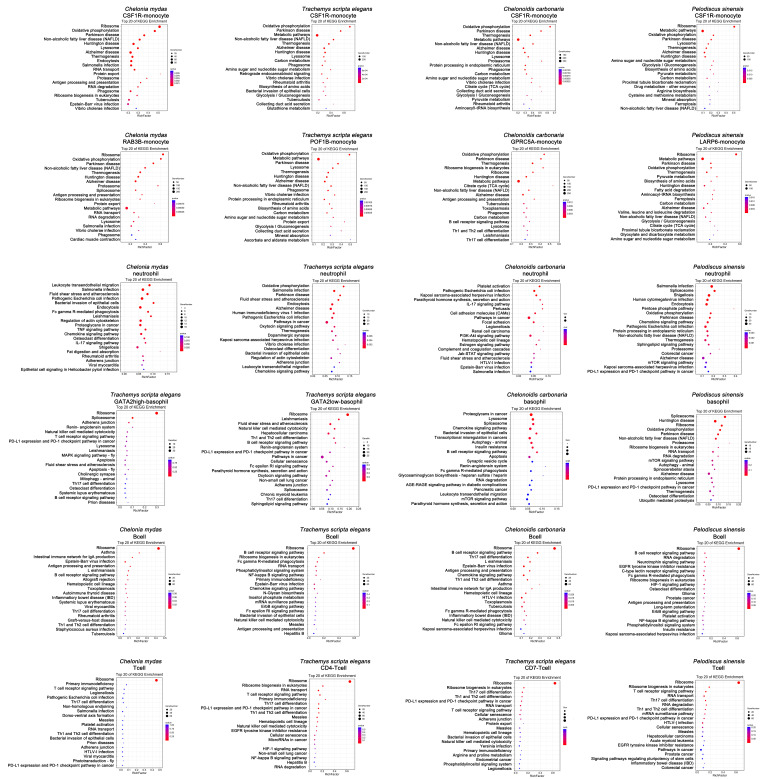
KEGG enrichment analysis of monocytes, neutrophils, basophils, B cells, and T cells in testudines.

**Figure 6 cells-11-04023-f006:**
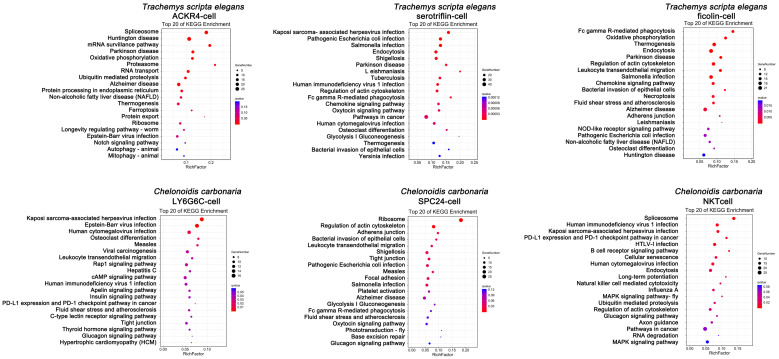
KEGG enrichment analysis of newly identified peripheral immune cells in testudines.

**Figure 7 cells-11-04023-f007:**
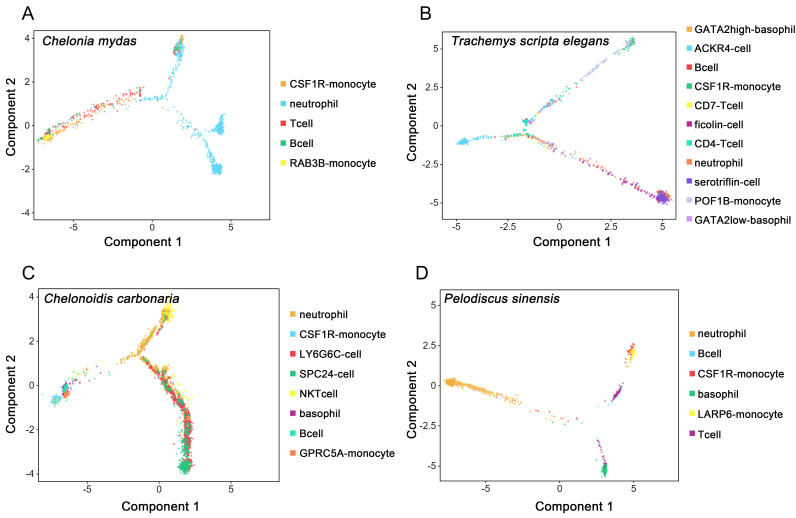
Differentiation trajectories of peripheral immune cells in testudines. (**A**) *C. mydas*; (**B**) *Trachemys scripta elegans*; (**C**) *C. carbonaria*; and (**D**) *P. sinensis*.

## Data Availability

The data presented in this study are available in Appendix A.

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
