# Peer review of "Comparison of the Single-Cell Immune Landscape of Testudines from Different Habitats"

_cells, 2022, doi:10.3390/cells11244023_

Round 1

Reviewer 1 Report

Dear editor,

Immunological adaptation is an important challenge for testudines to respond to the dramatic changes of the environmental pathogens during their diversification. In this study, the authors determined and compared the composition, function, and differentiation trajectory of peripheral immune cells in four representatives of testudines from different habitats. I think it is very interesting and will make an important contribution to the understanding of the immunological adaptation mechanism of testudines to different habitats.

Suggestions:

Section Introduction

1. Line 53, change ‘to study the classification of turtles’ to ‘to study the classification of the blood cells of turtles’. Please check whether the change is correct. The previous presentation might be misunderstood as the taxonomy of turtles.

2. More, please give more explanation to the different classification of the blood cells using different methods in previous studies.

3. Regularly, the aims or the focus of the paper were given in the last paragraph of the Introduction.

4. Why you choose the four representatives of testudines for this study? Please explain it.

Section Materials and Methods

5. 2.1 Sample collection, please give the details of the habitats of the four turtles you sampled.

6. Is any sort of chemical or clinical process has been done or not in this study. If yes, please provide a statement confirming the study was carried out in compliance with the ARRIVE guidelines in the declaration section.

Section Results

7. Please check the orders of the figures, I think the Fig.4 should be Fig.3, etc.

8. Could the quality of figures (especially Fig. 5) be improved, some information on the figures is unclear.

Section Discussion

9. Line 300, change ‘up-regrated’ to ‘up-regulated’.

10. It is interesting to find the different composition and function of the immune cells of the four turtle representatives from different habitats. Please give more explanations to the difference in the Section Discussion if possible.

Author Response

Dear reviewer #1,

Thank you for spending your precious time to review our manuscript, and thank you for your affirmation as well as your constructive and insightful comments on our manuscript. We have added the details of material and method and discussion in the revised manuscript. In addition, we have addressed all the comments one by one and listed as follows. After a systematical revision according to your comments, we know that our manuscript having obvious improvements, thank you again, and we hope these changes would make the manuscript suitable for publication.

Question 1: Line 53, change ‘to study the classification of turtles’ to ‘to study the classification of the blood cells of turtles’. Please check whether the change is correct. The previous presentation might be misunderstood as the taxonomy of turtles.

Answer: Thank you for your kind correction. We have changed it in the revised manuscript (line 53).

Question 2: More, please give more explanation to the different classification of the blood cells using different methods in previous studies.

Answer: Thank you very much for your kind advice. More information on different classification of the blood cells using different methods in previous studies has been supplemented in the introduction of the revised manuscript (line 54-58).

Question 3: Regularly, the aims or the focus of the paper were given in the last paragraph of the Introduction.

Answer: Thank you for your kind remind. We have changed the order in the introduction of the revised manuscript (line 84-88).

Question 4: Why you choose the four representatives of testudines for this study? Please explain it.

Answer: Thank you for your great question and give us a chance to classify it here. There are two reasons for choosing these four representative testudines in this study: 1) Their genome-wide have been published in the database; 2) Chelonia mydas, Trachemys scripta elegans, Chelonoidis carbonaria, and Pelodiscus sinensis are representative species in marine, freshwater, terrestrial, and sediment habitats, respectively.

Question 5: 2.1 Sample collection, please give the details of the habitats of the four turtles you sampled.

Answer: Thank you for your kind suggestion. The details of the habitats of the four turtles have been supplemented in the revised manuscript (line 91-96).

Question 6: Is any sort of chemical or clinical process has been done or not in this study. If yes, please provide a statement confirming the study was carried out in compliance with the ARRIVE guidelines in the declaration section.

Answer: Thanks for your kind remind. There is not any sort of chemical or clinical process in this study.

Question 7: Please check the orders of the figures, I think the Fig.4 should be Fig.3, etc.

Answer: Thank you for your kind remind. We have changed in the revised manuscript (line 184 and 211).

Question 8: Could the quality of figures (especially Fig. 5) be improved, some information on the figures is unclear.

Answer: Thank you for your kind suggestion and we spent plenty of time on improving the quality of figures which is crucial for the readability of the manuscript. We have increased font size and improved resolution and clarity of figures, after these works, we find that no problem to see every figure in details clearly if we enlarge pdf/word page when we are reading the manuscript. Of course, the authors also thought that extracting all figures from the manuscript and putting them by enlarged figures as an appendix at the end of this paper, but that would add many pages, think twice, we choose to improve the resolution and clarity of figures. After the improvement of figures resolution, there isn’t problem for the readability of all figures, we hope that the quality of all figures can satisfy the requirements of the Journal.

Question 9:. Line 300, change ‘up-regrated’ to ‘up-regulated’.

Answer: Thank you for your kind remind. We have changed it in the revised manuscript (line 314).

Question 10: It is interesting to find the different composition and function of the immune cells of the four turtle representatives from different habitats. Please give more explanations to the difference in the Section Discussion if possible.

Answer: Thank you for your great suggestion. More information has been supplemented in the revised manuscript (line 477-482).

Reviewer 2 Report

The author illustrated the peripheral immune landscape of testudines (Chelonia mydas, Trachemys scripta elegans, Chelonoidis carbonaria, and Pelodiscus sinensis) from different habitats using single-cell RNA sequencing (scRNA-seq) technique. There are some points that need to be improved.

Major comments:

1.     The readability of figures should be improved.

2.     The results of quality control are not showed

3.     The methods are not described clearly.

First, the method of developmental trajectory is not mentioned;

Second, Why the threshold of Log2FC is 0.36;

Third, based on the function FindClusters in package Seurat, different resolutions will produce different numbers of cell clusters, Which and why the ‘resolution’ is selected in the study.

4.     The results of DEGs among different clusters are not supplied. The presentation of marker genes is insufficient.

5.     The explanation of developmental trajectory should be improved.

Author Response

Dear reviewer #2,

Thank you very much for your careful review on our manuscript. We carefully studied your feedback and suggestions on our manuscript, and we have revised them according to your constructive comments. We hope these changes would make the manuscript suitable for publication.

Question 1: The readability of figures should be improved.

Answer: Thank you for your kind suggestion and we spent plenty of time on improving the quality of figures which is crucial for the readability of the manuscript. We have increased font size and improved resolution and clarity of figures, after these works, we find that no problem to see every figure in details clearly if we enlarge pdf/word page when we are reading the manuscript. Of course, the authors also thought that extracting all figures from the manuscript and putting them by enlarged figures as an appendix at the end of this paper, but that would add many pages, think twice, we choose to improve the resolution and clarity of figures. After the improvement of figures resolution, there isn’t problem for the readability of all figures, we hope that the quality of all figures can satisfy the requirements of the Journal.

Question 2: The results of quality control are not showed.

Answer: Thank you for your kind remind. The results of quality control have been supplemented in the supplementary material (Supplementary table 3) and revised manuscript (line 162).

Question 3: The methods are not described clearly. First, the method of developmental trajectory is not mentioned; Second, Why the threshold of Log2FC is 0.36; Third, based on the function Find Clusters in package Seurat, different resolutions will produce different numbers of cell clusters, Which and why the ‘resolution’ is selected in the study.

Answer: Thank you for your great suggestion which is very important for helping us to improve our manuscript. First, the method of developmental trajectory has been supplemented in the revised manuscript (line 153-156). Second, the screening criteria of differential gene in single-cell RNA sequencing is more relaxed than that of ordinary RNA sequencing; Meanwhile, log2 FC ≥ 0.36 was the default parameter of that Seurat screening up-regulated genes of clusters, it is also common screening criteria, such as shrimp immune cells (Yang et al., 2022). Third, we agree with you very much that different resolutions will produce different numbers of cell clusters. We tried different resolution, and found the identity of each cell cluster could be better identified when the resolution was 0.6. Therefore, the resolution of 0.6 is used in this study. We have applied it in the revised manuscript (line 141-142).

REFERENCES

Yang P, Chen Y, Huang Z, et al. Single-cell RNA sequencing analysis of shrimp immune cells identifies macrophage-like phagocytes[J]. Elife, 2022, 11: e80127.

Question 4: The results of DEGs among different clusters are not supplied. The presentation of marker genes is insufficient.

Answer: Thank you for your kind comments. We have supplied the results of DEGs among different clusters in the supplementary material (Supplementary file 1) and revised manuscript (line 163-165).

Question 5: The explanation of developmental trajectory should be improved.

Answer: Thank you for your kind suggestion. We have improved the explanation of development trajectory in the discussion of revised manuscript (line 453-463).

Reviewer 3 Report

General comments:

In the present manuscript, the authors investigated the immune cell composition and function of peripheral blood immune cells from four testudines species. This study is in my opinion very relevant to better understand diversity among species, their adaptation to different biological niches, and, therefore, improve classification of the species.

The manuscript is well written, and the data is presented concisely.

The rationale for the study is clearly presented and well explained.

All gene names should be written in italic style to not be confounded with proteins.

Were the turtle transcriptomes compared to any reference data, and if so, which data set?

Specific comments:

Introduction

Line 83: “The lifespan of wild P. sinensis is over 60 years old”, please change to: “The  lifespan of wild P. sinensis is over 60 years.”

Methods

Line 103/ 104: tube containing a small amount of EDTA-K2 small amount of EDTA-K2, please specify the amount

Results:

3.1. What was the rationale for the selection of cell type specific marker genes? Was your selection based on existing data in the literature? Or did you refer to marker genes from other species or animals? Please explain in the text and/or provide references.

Fig. 4. The font size of heat map needs to be increased, the gene names and legends are difficult to read

Fig.3./Fig.4.: The sequence of the figures needs to be corrected. Here, Fig.4. comes before Fig.3.

Fig.3. (UMAP): The color coding of the cell types should be uniform among the figures, this makes it easier for the reader to identify the cell cluster.

 Fig. 5. Please increase the font size of the graphs. It´s impossible to read any annotation.

 Line 259: … newly discovered innate immunity cell types…, should be replaced by: newly discovered innate immune cell types

Fig. 6. Please increase the font size of the graphs.

Line 273: B cells were essential components…, replace by: B cells are….

Line 282: T cells were essential cells in the immune system that were designed…, please improve, as for example. : T cells are essential for maintaining the immune balance by supporting the protection from foreign pathogens, suppressing excessive immune reactions, and preventing autoimmunity.

Author Response

Dear reviewer #3,

Thank you very much for your thoughtful comments on our manuscript. Your suggestions make the paper more perfect. Now we have revised it according to your constructive comments and every revision was marked in red in the revised manuscript.

Question 1: All gene names should be written in italic style to not be confounded with proteins.

Answer: Thanks very much. According to your suggestion, all gene names are written in italic style.

Question 2: Were the turtle transcriptomes compared to any reference data, and if so, which data set?

Answer: Thanks for your kind suggestion. We have applied reference data in the revised manuscript (line 124-127).

Question 3: Line 83: “The lifespan of wild P. sinensis is over 60 years old”, please change to: “The lifespan of wild P. sinensis is over 60 years.”

Answer: Thanks for your kind suggestion. We have changed it in the revised manuscript (line 83).

Question 4: Line 103/ 104: tube containing a small amount of EDTA-K2 small amount of EDTA-K2, please specify the amount.

Answer: Thank you for your careful review. We have supplemented specify the amount in the revised manuscript (line 111).

Question 5: 3.1. What was the rationale for the selection of cell type specific marker genes? Was your selection based on existing data in the literature? Or did you refer to marker genes from other species or animals? Please explain in the text and/or provide references.

Answer: Thank you for your great questions. There are two rationales for the selection of cell type specific marker genes in our study: 1) the cell type specific marker genes were selected according to the database provided by Genedenovo Biotechnology Co., Ltd. (Guangzhou, China); 2) the cell type specific marker genes were selected according to the marker genes from other species or animals. In the discussion, we explained the rationale for the selection of cell type specific marker genes (line 376, 398, and 428). In addition, we also defined the genes specific highly expressed in cells of the four testudines as marker genes, such as MYOF, LAMP3.

Question 6: Fig. 4. The font size of heat map needs to be increased, the gene names and legends are difficult to read

Answer: Thank you for your kind suggestion. We have increased font size of figure in the revised manuscript (line 210).

Question 7: Fig.3./Fig.4.: The sequence of the figures needs to be corrected. Here, Fig.4. comes before Fig.3.

Answer: Thank you for your kind remind. We have corrected it in the revised manuscript (line 183 and 210).

Question 8: Fig.3. (UMAP): The color coding of the cell types should be uniform among the figures, this makes it easier for the reader to identify the cell cluster.

Answer: Thank you for your kind suggestion. We have adjusted the color coding of the cell types in the revised manuscript (line 210).

Question 9: Fig. 5. Please increase the font size of the graphs. It´s impossible to read any annotation.

Answer: Thank you for your kind suggestion. We have increased the font size of the graphs in the revised manuscript (line 238).

Question 10: Line 259: … newly discovered innate immunity cell types…, should be replaced by: newly discovered innate immune cell types.

Answer: Thanks for your kind suggestion. We have replaced it in the revised manuscript (line 276).

Question 11: Fig. 6. Please increase the font size of the graphs.

Answer: Thanks for your kind suggestion. We have increased the font size of the graphs in the revised manuscript (line 288).

Question 12: Line 273: B cells were essential components…, replace by: B cells are….

Answer: Thanks for your kind suggestion. We have replaced it in the revised manuscript (line 290).

Question 13: Line 282: T cells were essential cells in the immune system that were designed…, please improve, as for example. : T cells are essential for maintaining the immune balance by supporting the protection from foreign pathogens, suppressing excessive immune reactions, and preventing autoimmunity.

Answer: Thanks for your kind suggestion. We have improved in the revised manuscript (line 299-301).

Round 2

Reviewer 2 Report

The author illustrated the peripheral immune landscape of testudines(Chelonia mydas, Trachemys scripta elegans, Chelonoidis carbonaria, and Pelodiscus sinensis) from different habitats using single-cell RNA sequencing (scRNA-seq) technique. Most comments and suggestions have been solved.  There are still some points that need to be improved. 

1. the method of developmental trajectory, Monocle2, should be described in detail, such as , which genes used for pseudotime ordering.

2. Graphical illustration of experiment should be provided.

Author Response

Dear reviewer #2,

Thank you very much for your careful review on our manuscript. We carefully studied your feedback and suggestions on our manuscript, and we have revised them according to your constructive comments. We have added the details of developmental trajectory method and graphical illustration of experiment in the revised manuscript. We hope these changes would make the manuscript suitable for publication.

Question 1: The method of developmental trajectory, Monocle2, should be described in detail, such as, which genes used for pseudotime ordering.

Answer: Thank you for your great suggestion. The authors carefully studied your suggestion, and improved the description of developmental trajectory method (line 153-160). Thanks to your seriousness for help us improve our paper in details.

Question 2: Graphical illustration of experiment should be provided.

Answer: Thank you very much for your kind advice. We have added graphical illustration of experiment in the revised manuscript (line 213-217).